# Establishing an Intelligent Emotion Analysis System for Long-Term Care Application Based on LabVIEW

**Kai-Chao Yao** , **Wei-Tzer Huang \*** , **Teng-Yu Chen, Cheng-Chun Wu and Wei-Sho Ho \***

Department of Industrial Education and Technology, National Changhua University of Education, No. 1, Jin-De Rd., Changhua 500, Taiwan; kcyao@cc.ncue.edu.tw (K.-C.Y.); m1031022@gm.ncue.edu.tw (T.-Y.C.); m1031023@gm.ncue.edu.tw (C.-C.W.)

\* Correspondence: vichuang@cc.ncue.edu.tw (W.-T.H.); d0931001@gm.ncue.edu.tw (W.-S.H.)

**Abstract:** In this study, the authors implemented an intelligent long-term care system based on deep learning techniques, using an AI model that can be integrated with the Lab's Virtual Instrumentation Engineering Workbench (LabVIEW) application for sentiment analysis. The input data collected is a database of numerous facial features and environmental variables that have been processed and analyzed; the output decisions are the corresponding controls for sentiment analysis and prediction. Convolutional neural network (CNN) is used to deal with the complex process of deep learning. After the convolutional layer simplifies the processing of the image matrix, the results are computed by the fully connected layer. Furthermore, the Multilayer Perceptron (MLP) model embedded in LabVIEW is constructed for numerical transformation, analysis, and predictive control; it predicts the corresponding control of emotional and environmental variables. Moreover, LabVIEW is used to design sensor components, data displays, and control interfaces. Remote sensing and control is achieved by using LabVIEW's built-in web publishing tools.

**Keywords:** long-term care; LabVIEW; convolutional neural network (CNN); emotional analysis

## 1. Introduction

According to the United Nations World Population Prospects 2019 report, the total global population in 2020 will be close to 7.8 billion people, of which 730 million people are over 65 years old, accounting for about 9.3%. It is estimated that by 2050, the number of elderly people will reach 1.5 billion, doubling the growth rate from 2020, accounting for about 16%, showing that the aging of the global population is a common phenomenon [1]. In recent years, the world has faced the double attack of chronic diseases and emerging infectious diseases. Chronic diseases that often occur in the elderly consume a lot of medical resources, and the severe special infectious pneumonia (COVID-19) pandemic has a serious impact on human mental health, showing that chronic diseases and mental diseases have become common problems in modern society. The Industrial Technology Research Institute and Nan Shan Life jointly proposed a white paper on long-term care for the elderly in Taiwan in 2021, indicating that the elderly industry will combine artificial intelligence and machine learning in the future to transform traditional care into more high-quality and efficient smart care [2]. According to a survey conducted by Common Wealth Magazine in 2021, the average life expectancy of Taiwan's rural residents is 7 years shorter than that of urban residents, and the lack of medical networks and long-term care is the primary reason. Therefore, it is one of the motives of this study to bridge the gap [3]. The development of a remote care system has also become an important issue in long-term care for the elderly [4].

The goal of this research is to construct an AI emotion recognition model and a long-term care medical system with remote monitoring functions. The physical and mental health of the elderly is extremely valued. While gradually losing physical strength, the elderly must maintain a positive mental state to reduce the impact of aging. The mental

state can be judged by observing the face of the elderly [5–7]. In [8], the authors proposed to test the accuracy of successfully predicting seven facial micro-expressions, i.e., happy, sad, angry, scared, surprised, and deceitful-using facial expressions for the real-time temptation and aversion recognition dataset (FER-2013), which is the 2013 Facial Expression Recognition Dataset (FER-2013) provided by Kaggle and presented at the In-2013 International Conference on Machine Learning (ICML) [9]. In the literature [10], seven categorical feature points of emotional microemotic expressions are mentioned. Through real-time analysis of the facial emotions of the elders, the mental state of elderly can be judged and learned, such as anger, depression, and sadness [11]. These concepts can feedback on medical care to assist long-term care and medical care staff in decision-making for helping [12].

In modern society, the advancement of AI allows humans to save more time and resources and make life more convenient. For example, in [13], a lifelike virtual speaker is built through deep learning technology, which, in addition to a beautiful static appearance, uses AI technology to simulate mouth movements, facial expressions, and body movements to truly synchronize with sound. Deep learning techniques have been very successful in various fields, even in the medical field [14–19]. Due to the aging population, the issue of long-term care for the elderly, which has gradually attracted attention, has also attracted the attention of researchers. It is hoped that emerging technologies can solve the existing difficulties of long-term care for the elderly. In the literature [20], the application of convolutional neural networks to Alzheimer's disease (AD) for common diseases in the elderly is explored in diabetes and nursing. Among them, the analysis results of the CNN model are better than those of the full convolutional neural network (FCNN) and the partition effect of the support vector machine (SVM) algorithm.

This study proposes to develop an AI model for facial recognition, so that the care system can possess the ability of facial emotion recognition, which can be applied to the long-term care needs of the elderly. This developed system uses LabVIEW to collect remote information, such as monitoring images, temperature, humidity, etc. In the program design, Python was used to achieve the artificial neural network of CNN, and a facial recognition AI model was built inside LabVIEW to predict the emotions of the care recipients, and then LabVIEW gave the appropriate corresponding care needs according to the judgment of the AI model.

The use of LabVIEW software enables functions such as data acquisition, storage, retrieval, display and control [21], and accomplish wireless transmission and real-time image monitoring [22]. The intelligent model is used to judge the facial emotional changes of the care recipients in the long-term care environment, and then output control signals. LabVIEW uses built-in signal output modules and network publishing tools to build a remote sensing and control function, which enables the system to monitor the long-term care environment and each sensor transmit the sensing data through a wireless connection during the monitoring, also enables the system to use internet access devices in different locations for real-time monitoring and control.

In addition, in the system deployment of the control side, the output control signal can be used to activate different relays to provide different care needs such as alarm, air conditioning, lighting, sound and sending electronic signals [23]. This research integrates LabVIEW and IoT technology and achieves the construction of an intelligent home-based long-term care system.

The facial emotion recognition system has a wide range of applications in the fields of smart home and medical care. Functionally, it needs to be able to accurately recognize the face or emotion to reflect the correct judgment output [8,23–34], but in the above articles, there are no systems built by integrating LabVIEW and Python. In [35–42], different recognition applications and computing methods are mentioned, such as [42] proposes CNN architecture to segregate different plant images from the sequences collected. In this research, convolutional cellular neural networks (CNN) is utilized to achieve identification and judgment ability in the system, this technology is one of the more mature technologies in intelligent image detection [43].

## 2. Preliminaries

CNNs have very important applications in different fields [43]. The CNN method is an extension of the multilayer perceptron (MLP) method for two-dimensional processing of data, with the most common applications being image processing and feature recognition [44]. The difference between CNNs and MLPs is that each neuron in CNNs exists in two dimensions, while each neuron in MLPs has only one dimension. The most beneficial aspect of CNNs is reducing the number of parameters in ANNs. This achievement has prompted both researchers and developers to approach larger models in order to solve complex tasks, which was not possible with classic ANNs. The most important assumption about problems that are solved by CNNs should not have features which are spatially dependent. In other words, for example, in a face detection application, we do not need to pay attention to where the faces are located in the images [45]. Therefore, this research builds an AI model for real-time image sentiment analysis based on the CNN model. The structure diagram is shown in Figure 1.

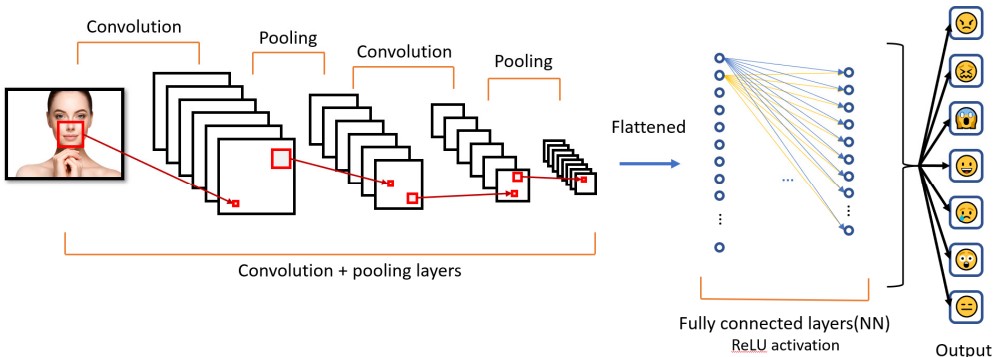

**Figure 1.** Structure diagram of CNN convolutional neural network.

CNNs' convolutional neural network consists of three layers, namely convolution operation (CONV layer), pooling operation (Pooling layer) and fully connected layer.

A convolutional layer is a layer that contains an entire set of filters, each of which is convolved with the input image. At the beginning of the operation, all the convolution kernels are randomly initialized. Then, the neural network will calculate the neuron output according to the coefficients of the convolution kernel and connect the calculated neuron weights to the input of the next convolution layer.

RELU (Rectified Linear Unit, linear rectification function) is a kind of activation function in a neural network. The main purpose of the activation function is to increase the nonlinearity of the neural-like network model, so that the defined neural-like network can be more active and learn, and avoid being rigid like a linear function.

Pooling layers are used to reduce the dimensions of the feature maps. Thus, it reduces the number of parameters to learn and the amount of computation performed in the network. The pooling layer summarizes the features present in a region of the feature map generated by a convolution layer [46].

CNN shares the best features that possess local interconnection characteristics making it to be easily realizable for the very large-scale integration (VLSI) implementation either as planar or as multilayer structures and real-time continuous and high-speed parallel signal processing feature. By ordering the cells, in the case where cells are linear interactions and the input of each cell is constant, the state equation of CNN is described by the first-order nonlinear differential system shown in Equation (1) [47].

$$\dot{x}(t) = -\mathrm{R}x(t) + \mathrm{A}y(t) + \mathrm{u} \tag{1}$$

## 3. System Structure

LabVIEW is the core main programming software of the graphical user interface (GUI). In addition to providing human-machine interface presentation and data reception, it can

also develop the system on the web page or develop application programs in the form of API, remote monitoring, CAM image acquisition, and corresponding controls. Python is used to build a deep learning AI model embedded in LabVIEW. This research is developed on the basis of a common and mature CNN model. However, the supervised learning method of manual labels is time-consuming and prone to overfitting during the research process. Therefore, in the future, the unsupervised Generative Adversarial Network (GAN) will be imported to conduct research on the semi-supervised method of confrontation generation with this model. Figure 2 shows the system structure of the data, sentiment analysis and data processing of the intelligent long-term lighting environment. In addition, LabVIEW stores the newly collected image files in a new database for AI model calculation on the server. Currently, it will also be set to the cloud storage method. In the future, it can directly perform cloud edge computing to speed up data processing and transmission and reduce delays. On the interface, NI DAQ Card will be used to achieve practical sensing and control capabilities. In addition, new images captured in real-time while the system is running will be the source of data for continuous training and validation of the model.

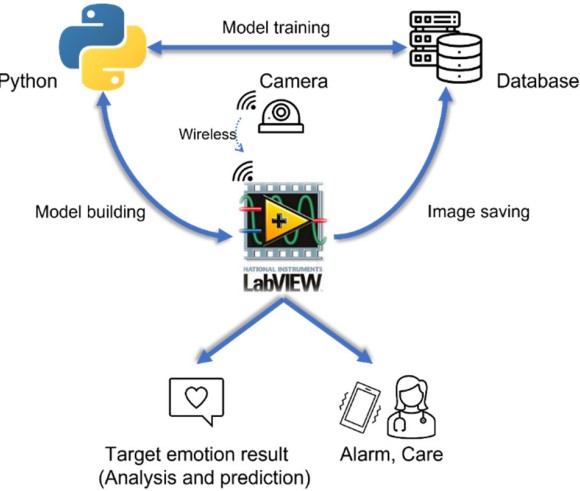

**Figure 2.** The system architecture of Smart long-term care.

Once the system is completed, the function of micro-expression emotion analysis of the system module can be integrated into the long-term care system, such as the architecture of the smart long-term care monitoring system shown in Figure 3, which shows the corresponding control deployment in the right side, such as notifying the caregiver, adjusting the environment, or transmitting warning signals, etc.

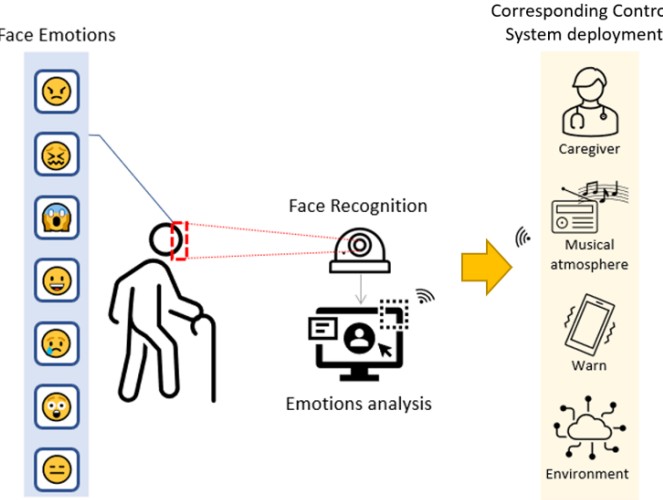

**Figure 3.** Structure of Smart Long-term care system.

In the emotion recognition system, after the image captures the facial features, the points of difference are distinguished from them. For example, eyebrow contour, eye contour, pupil, nostrils, mouth contour, and mouth center are organized into seven kinds of micro-expression categories, which have distinct classification criteria. The classification criteria are:

1. happy: the cheek muscles rise, the corners of the mouth rise, the corners of the mouth pull back, the eyebrows are flat, and the eyes become smaller;
2. sadness: upper eyelid drooping, dilated pupils, corners of the mouth pulling down, cheeks pulling down, eyebrows locked deeply;
3. anger: enlarged nostrils, enlarged eyes;
4. disgust: raise your nose and raise your mouth;
5. fear: the middle of the eyebrows is crowded together;
6. surprise: the mouth is slightly open, the pupils are dilated, and the eyebrows are raised;
7. contempt: The corners of the lips tighten and lift only one side of the face, and one eyebrow rises;

From Figure 4, the same emotion under the feature points and age are not much different, and after the image is grayscale processed, the skin color will not have a big impact, and the AI model can be trained to identify according to the feature points of these emotions.

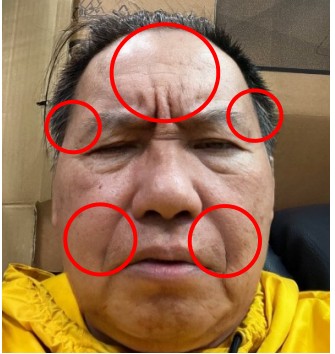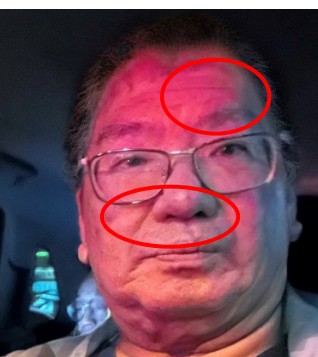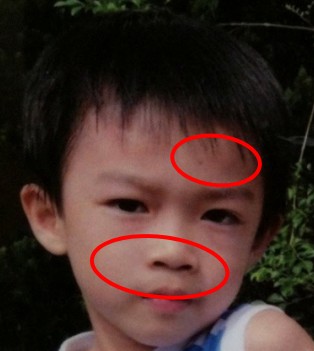

**Figure 4.** The image on the left is an example of a sad feature map; in the middle and to the right are example diagrams of the characteristics of the concept of different ages.

## 4. Main Results

In this study, LabVIEW is used to construct a monitoring-control interface for the AI model that integrates facial feature recognition and emotion analysis. The monitoring and control interface is shown in Figure 5. The human-machine interface design includes (A) environment monitoring block, (B) user login block, (C) block of emotion prediction results, (D) block of instant emotion analysis of facial expressions, and (E) AI model scheduling control area. In addition, the established model archives will be scheduled in the system with the LabVIEW tool library, Python nodes, which are used to connect the trained and tested AI models to predict the actual control needs in the long-term care environment. Using this method of construction, we can build AI models into neural networks that can predict emotions and perform intelligent control based by LabVIEW. In addition, LabVIEW features can help create friendly GUIs that facilitate neural network operation, analysis, and monitoring.

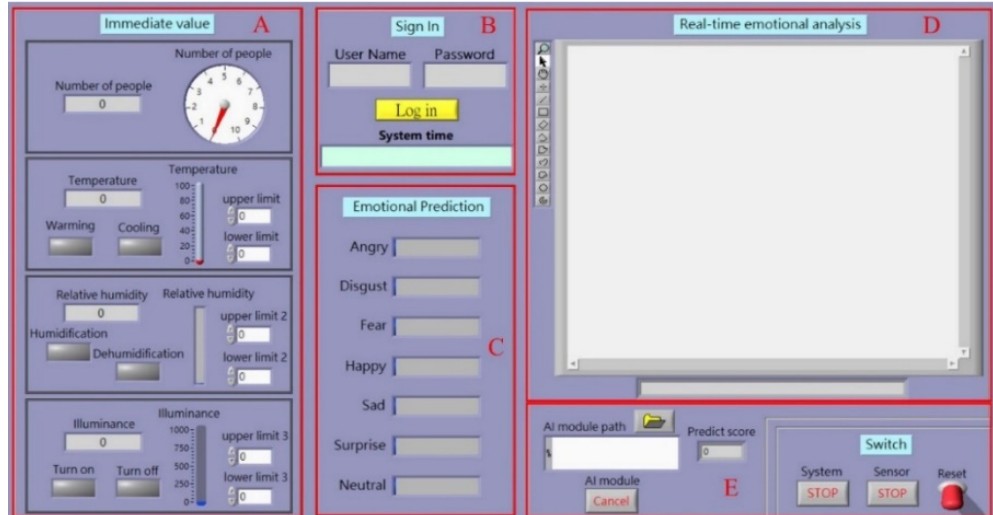

**Figure 5.** Monitoring and control interface for emotional analysis of long-term care. (**A**) environment monitoring block; (**B**) user login block; (**C**) block of emotion prediction results; (**D**) block of instant emotion analysis of facial expressions; (**E**) AI model scheduling control.

The proposed system is designed to be able to integrate various AI models in future developments. In addition to the CNN model trained in this paper, other deep learning models are used for future updates or applications. These AI models can be selected in the human-machine Interface block E. In this study, the selected AI model is a CNN with a predictive control MLP framework. Block E can show the predicted quantization scores, reset data, stop detection, and select AI models. Figure 6 shows the situation when different AI models are selected.

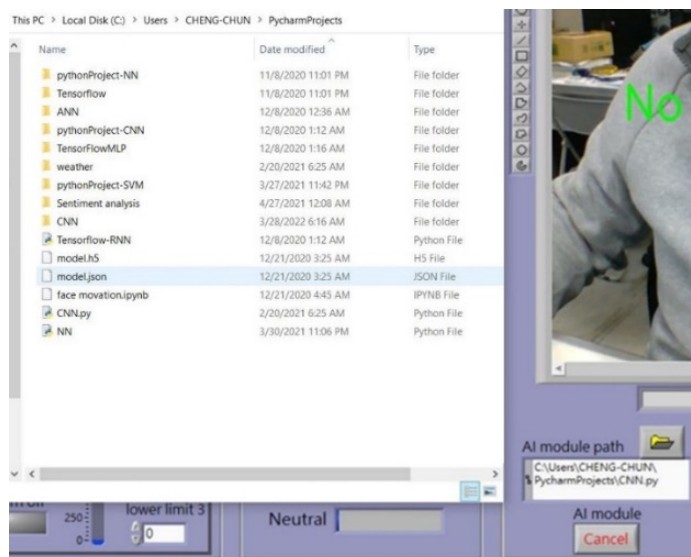

**Figure 6.** HMI block E selects various AI Model.

## 4.1. Environmental Monitoring

In the environmental monitoring section, the HMI displays deployed sensor data and detected real-time environmental condition data. The data displayed is real-time data captured by the sensor, and converted to standard units by LabVIEW calculations, including the current number of people, current temperature, humidity, and brightness. Figure 7 shows the program design diagram of environmental monitoring. The functional design is set to monitor environmental conditions to further predict and analyze the control

of the environment. Since the environment may be one of the factors that affect the target person's emotions, the module collects environmental data: crowd density, temperature, light, etc., limited by the current lack of environmental data and emotional data. In the future, through the collection of these data and this research, the analysis of sentiment analysis can further explore correlations and can be planned to be added as control factors to make the system more complete.

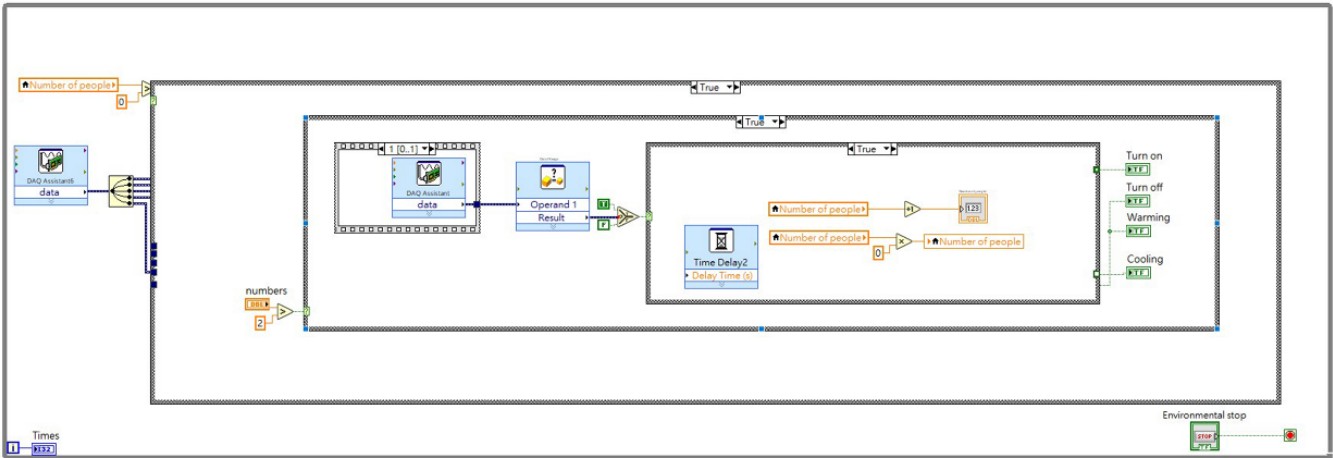

**Figure 7.** The program diagram pertaining to environmental monitoring and control.

*4.2. Sign-in Function*

In addition to protecting the system from illegal people and protecting privacy, the system records who logged in and when to a file for future research and tracking. Authentication is deployed before the operating system, as shown in block B of Figure 5. At this stage, it is possible to prevent the elderly from accidentally touching, but for many inconvenient factors such as operations, after the model verification is completed, it is integrated with face recognition as a safety mechanism. If the login process fails, you cannot enter the monitoring system. Figure 8 depicts an image of a successful system login.

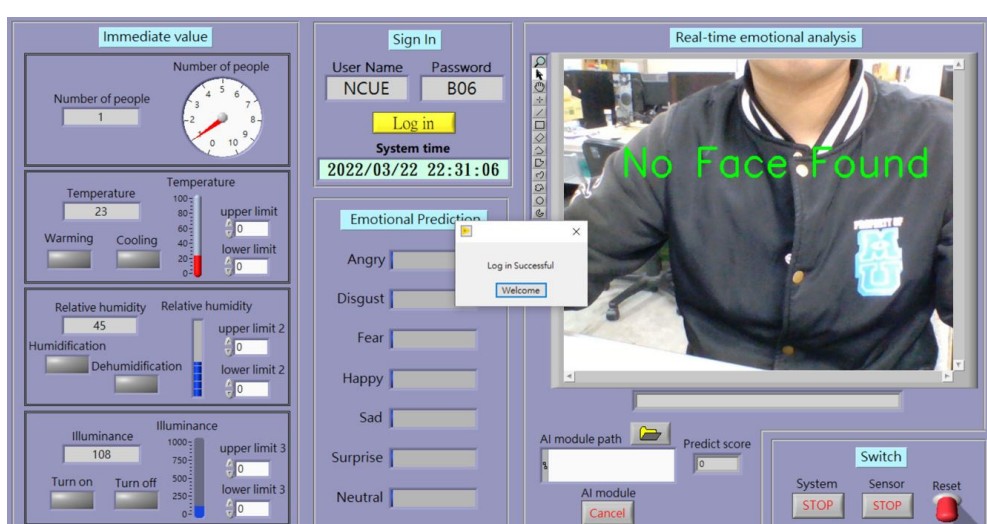

**Figure 8.** System login situation.

*4.3. Sentiment Prediction*

In the AI model design, Python is used to construct the required AI modules, train and test them first, and then verify the feasibility of the model. The part using the programming approach is shown in Figure 9. The AI model (Python CNN architecture) is embedded in

the system as an API using the LabVIEW Python node. The system receives the detected images instantly on the human-machine interface, and the trained model is used to verify the facial expressions and perform predictive analysis at the back-end of the system. In addition, the detected image data is further transferred back to the new training set for subsequent model validation. The obtained expressions are quantified by LabVIEW for the prediction scores of the sentiment, as shown in Figure 10. This section corresponds to A in Block Figure 5, and the results of these operations will be used for various control decisions in the long-term care environment.

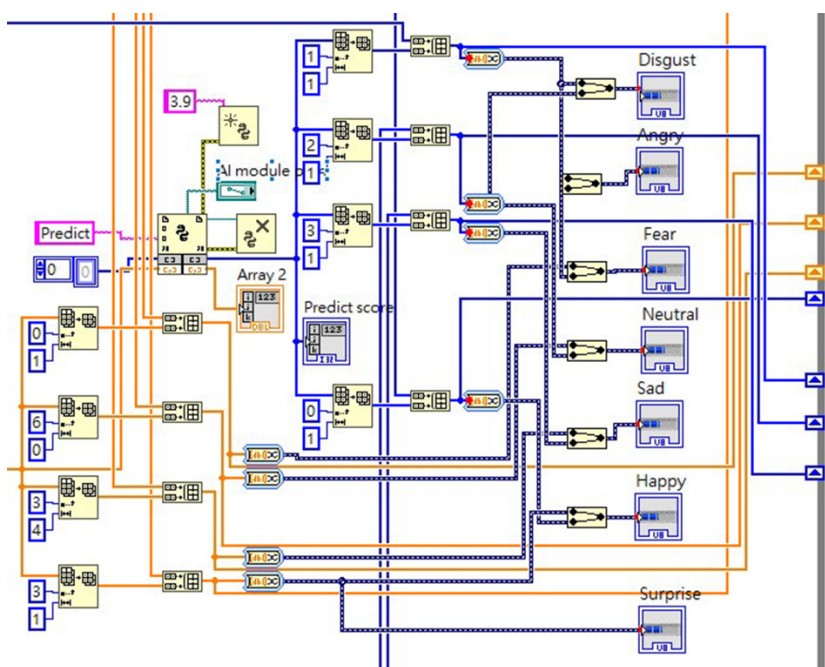

**Figure 9.** The emotional prediction program diagram of Python model imported into LabVIEW calculation.

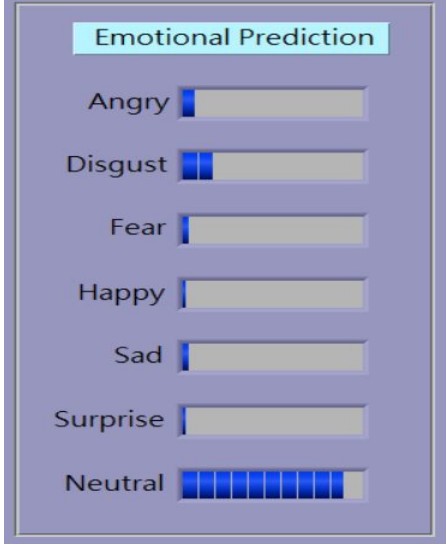

**Figure 10.** Quantitative results of sentiment prediction.

### 4.4. Facial Expression Image Monitoring

Figure 11 shows the predicted value of emotion for real-time face recognition or tells the system if the face image is detected by CAM. The predicted emotion will be analyzed to further control the environment or to alert and inform the medical staff for the next

strategy. Due to the current difficulties in obtaining national laws and personal data, the strategy section is still inconclusive, and if the identity of the person can be detected in the future, the medical staff can tailor it to the target. Figure 12 shows part of the procedure for real-time facial expression monitoring and emotional recognition analysis.

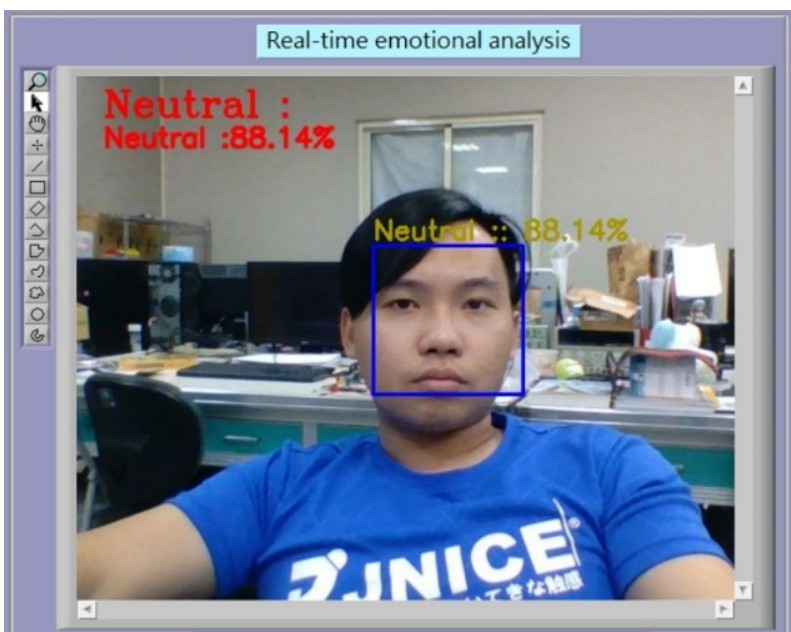

**Figure 11.** Real-time facial expression monitoring and emotion analysis display.

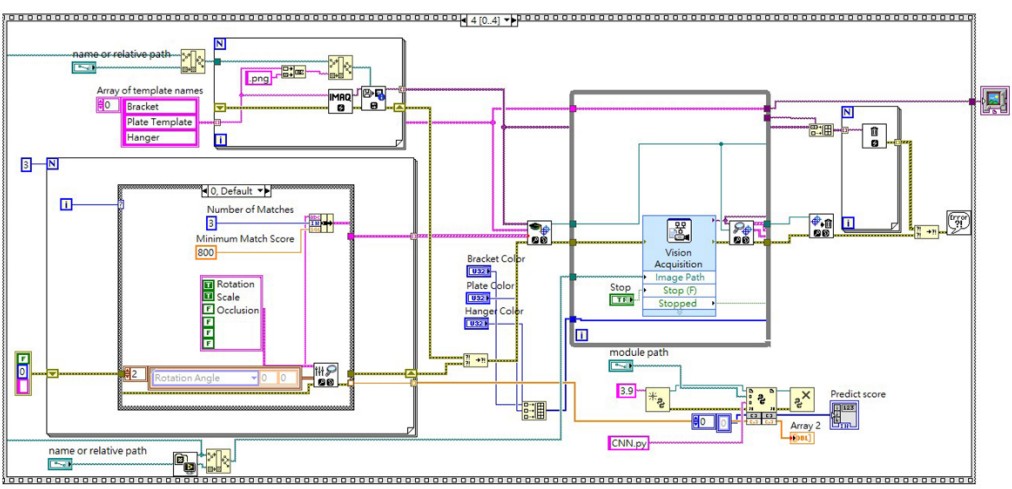

**Figure 12.** Partial program diagram of facial expression real-time monitoring and emotion recognition analysis.

### 4.5. Model Construction and Methods

In this study, we used deep learning techniques to predict emotional changes in facial expressions. Deep learning is a branch of machine learning, a type of machine learning that is repeated in training and testing to simulate deeper and more complex human thinking such as neural network. In this research, CNN convolutional neural network structure is used shown in Figure 1. The captured facial expression pictures will be performed neural network-like operations to achieve image feature extraction, analysis operations, and prediction different facial expressions. Figure 13 shows the AI model building steps of this research. The steps are as follows:

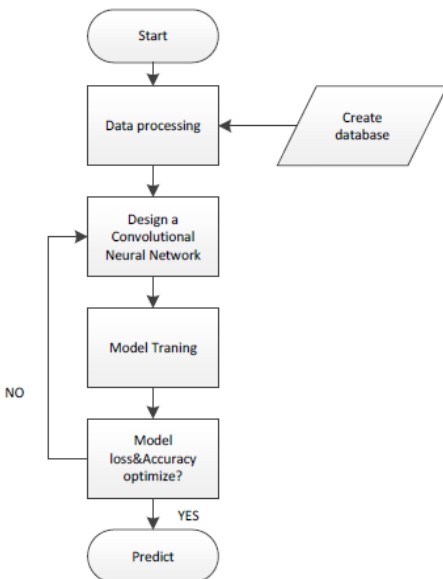

**Figure 13.** Flow chart of design steps of the learning model.

Step 1: Collect the images to be trained, convert the images into numerical expression and label them as training data in the database. In this study, the data classification of the FER-2013 dataset was referred to, and the dataset was labeled according to seven different classifications of happy, anger, sadness (pain), fear, disgust, surprise, and contempt, and the number of data in the dataset is shown in Table 1.

**Table 1.** Statistical Table of Testing and Validation Data of Research Datasets.

| Micro-Expression | Validation Data | | | Training Data | | |
|---|---|---|---|---|---|---|
| (Classification) | Elder | Others | Total | Elder | Others | Others |
| happy | 345 | 1248 | 1593 | 2315 | 6784 | 9099 |
| anger | 244 | 976 | 1220 | 1042 | 4253 | 5295 |
| sadness | 287 | 881 | 1168 | 2151 | 5382 | 7533 |
| fear | 193 | 829 | 1022 | 1097 | 4721 | 5818 |
| disgust | 43 | 153 | 196 | 282 | 805 | 1087 |
| surprise | 218 | 716 | 934 | 1126 | 3686 | 4812 |
| contempt | 284 | 996 | 1280 | 1972 | 5819 | 7791 |
| | 1614 | 5799 | 7413 | 9985 | 31,450 | 41,435 |

Step 2: Use Python to build the convolution layer and pooling layer of the neural network, which includes 4 times of non-linear convolution operation of the convolution layer and 4 times of pooling process as shown in Figure 14, next, connect the fully connected layer at the back end, and output the classification result such as shown in Figure 15.

```
#Creative model structure
model = Sequential()
#The first convolutional layer
model.add(Conv2D(input_shape=(48,48,1),filters=32,kernel_size=3,padding='same',activation='relu'))
model.add(Conv2D(filters=32,kernel_size=3,padding='same',activation='relu')) #
model.add(MaxPool2D(pool_size=2, strides=2)) #half width and height
#The second convolutional layer
model.add(Conv2D(filters=64,kernel_size=3,padding='same',activation='relu'))
model.add(Conv2D(filters=64,kernel_size=3,padding='same',activation='relu'))
model.add(MaxPool2D(pool_size=2, strides=2))
#The third convolutional layer
model.add(Conv2D(filters=128,kernel_size=3,padding='same',activation='relu'))
model.add(Conv2D(filters=128,kernel_size=3,padding='same',activation='relu'))
model.add(MaxPool2D(pool_size=2, strides=2))
#The fourth convolutional layer
model.add(Conv2D(filters=256,kernel_size=3,padding='same',activation='relu'))
model.add(Conv2D(filters=256,kernel_size=3,padding='same',activation='relu'))
model.add(MaxPool2D(pool_size=2, strides=2))
```

**Figure 14.** convolutional layer and pooling layer construction process.

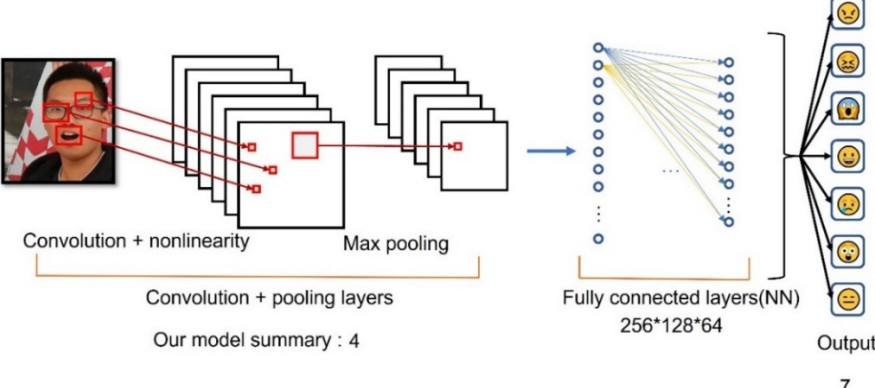

**Figure 15.** CNN structure design diagram of this study.

Step 3: Input the training data processed in Step 1 into the CNN-like neural network built in Step 2 for training.

Step 4: Analyze model loss and accuracy through training and test sets.

Step 5: After confirming that the model loss and accuracy have the recognition ability, the developed AI model will begin to perform real-time actual image recognition work.

Figure 15 is the architectural design diagram of the CNN AI model structure, and Figure 16 is the practical appearance diagram of the designed system that utilizes an NI-DAQmx to output control signals and control from a remote. The AI model begins by filtering, labeling, and processing selected images fetched from the CAM. Figure 17 is the numerical data and labels after image processing. Figure 18 shows the image test set before processing. These image label data will be provided to the AI model for learning and training. After repeated operations of four layers, in Figure 19, the detailed features of the pixels will be extracted, and finally used as the input of the fully connected layer and possessing the power to classify after the activation function with the fully connected layer, in Figure 20. Figure 18 shows the image test set before processing.

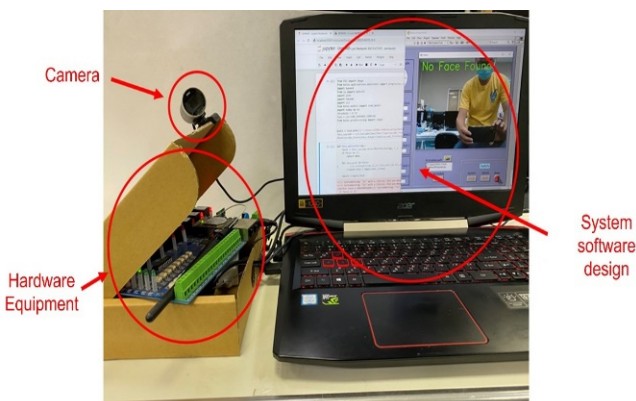

**Figure 16.** The appearance of the designed system in practice.

During the training process, Figure 19 shows the design of the AI model, which repeatedly performs convolution and pooling operations. Since the features after the convolution operation often cause wrong numerical image data transfer, the pooling operation provides another form of translation invariance, which continuously reduces the spatial size of the data, as shown in block A. The input image enters the first layer of convolution and pooling operation; the number of data and the amount of calculation will also decrease. This part can be seen in the list of output shapes in Figure 20. The image data decreases from $48 \times 48 \times 32$. This process will also reduce the probability of over-fitting because the pooling layer will remove inappropriate numerical data conversion operations

on the image caused by the AI model in the convolutional layer. This process can eliminate skew, stains, and deformations.

| Emotion | Pixels | Usage |
|---|---|---|
| 0 | 255 255 255 255 255 252 255 208 165 182 181 171 161 158 160 15·· | Training |
| 6 | 98 121 108 36 53 53 56 79 112 148 110 90 39 45 20 11 15 24 22 22·· | Training |
| 6 | 21 24 31 41 44 45 46 48 55 70 78 69 131 183 197 206 209 207 207··· | Training |
| 5 | 208 203 194 199 212 214 215 217 218 217 211 206 201 197 200 208· | Training |
| 6 | 232 230 236 188 133 136 135 136 136 139 142 147 153 154 159 164· | Training |
| 5 | 204 213 221 218 208 202 214 200 209 221 222 224 229 231 230 230· | Training |
| 3 | 142 158 186 209 190 169 192 171 153 163 144 42 35 48 64 77 87 10 | Training |
| 0 | 219 221 224 225 226 226 227 228 227 225 225 224 222 222 222 224· | Training |
| 5 | 255 255 255 254 255 236 167 149 150 137 148 135 122 132 128 118· | Training |
| 6 | 248 248 248 248 248 248 248 248 248 244 238 228 162 153 153 139· | Training |
| 0 | 62 58 55 48 48 48 48 58 66 68 76 88 97 118 125 119 110 104 109 1·· | Training |
| 2 | 132 139 144 145 145 144 147 141 124 105 116 102 92 119 129 97 90 | Training |
| 6 | 95 86 70 67 69 68 35 46 77 75 75 73 74 78 76 72 76 117 126 133 109 | Training |
| 6 | 87 70 59 62 84 88 89 55 36 41 89 116 54 51 64 74 60 55 57 100 103· | Training |
| 6 | 59 59 55 39 38 43 51 56 47 54 58 62 64 75 82 83 90 95 102 106 109· | Training |
| 2 | 94 84 83 80 82 79 77 77 75 77 114 162 173 154 148 150 159 168 165 | Training |
| 4 | 9 14 10 7 8 19 45 59 81 89 99 119 121 121 128 126 118 112 107 107· | Training |
| ⋮ | ⋮ | ⋮ |

**Figure 17.** The training model incorporates image processing numerical data and manual labels.

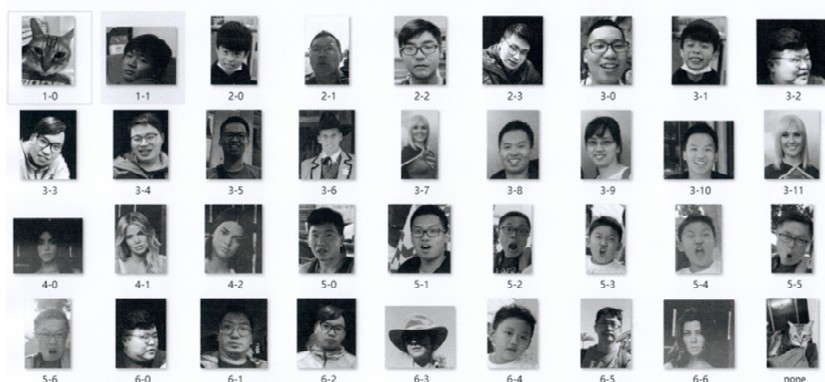

**Figure 18.** Input pre-training image captured by CAM.

```
Model: "sequential"
_________________________________________________________________
Layer (type)                 Output Shape              Param #
=================================================================
conv2d (Conv2D)              (None, 48, 48, 32)        320
_________________________________________________________________
conv2d_1 (Conv2D)            (None, 48, 48, 32)        9248
_________________________________________________________________
max_pooling2d (MaxPooling2D) (None, 24, 24, 32)        0
_________________________________________________________________
conv2d_2 (Conv2D)            (None, 24, 24, 64)        18496
_________________________________________________________________
conv2d_3 (Conv2D)            (None, 24, 24, 64)        36928
_________________________________________________________________
max_pooling2d_1 (MaxPooling2 (None, 12, 12, 64)        0
_________________________________________________________________
dropout (Dropout)            (None, 12, 12, 64)        0
_________________________________________________________________
conv2d_4 (Conv2D)            (None, 12, 12, 128)       73856
_________________________________________________________________
conv2d_5 (Conv2D)            (None, 12, 12, 128)       147584
_________________________________________________________________
max_pooling2d_2 (MaxPooling2 (None, 6, 6, 128)         0
_________________________________________________________________
dropout_1 (Dropout)          (None, 6, 6, 128)         0
_________________________________________________________________
conv2d_6 (Conv2D)            (None, 6, 6, 256)         295168
_________________________________________________________________
conv2d_7 (Conv2D)            (None, 6, 6, 256)         590080
_________________________________________________________________
max_pooling2d_3 (MaxPooling2 (None, 3, 3, 256)         0
_________________________________________________________________
dropout_2 (Dropout)          (None, 3, 3, 256)         0
```

**A**

**Figure 19.** The model design of convolution and pooling layer.

```
flatten (Flatten)           (None, 2304)              0
-------------------------------------------------------------
dense (Dense)               (None, 256)           590080
-------------------------------------------------------------
dropout_3 (Dropout)         (None, 256)               0
-------------------------------------------------------------
dense_1 (Dense)             (None, 128)            32896
-------------------------------------------------------------
dropout_4 (Dropout)         (None, 128)               0
-------------------------------------------------------------
dense_2 (Dense)             (None, 64)              8256
-------------------------------------------------------------
dropout_5 (Dropout)         (None, 64)                0
-------------------------------------------------------------
dense_3 (Dense)             (None, 7)                455
=============================================================
Total params: 1,803,367
Trainable params: 1,803,367
Non-trainable params: 0
```

**Figure 20.** The model design of flat layer and the fully connected layer.

Such as Figure 19 four-block red box, after the 4-layer convolution and pooling process designed in this study, the author refers to the FER-2013 database to convert the collected full-color images into a first-order matrix image, black and white is more friendly to the model. After the generalized convolution operation, such as Formula (2), the function x and y are measurable function defined on $R^n$. The convolution of x and y is denoted as $x * y$. It is the integral of the product of one of the functions after inversion and translation and the product of the other function, which is a pair A function of the amount of translation, the number of features obtained after the cumulative operation are displayed in the param column. The next stage, the flatten layer, will expand the feature data output from the convolution and pooling layer and performing dimension conversion for input data to the fully connected layer, and finally the AI model has the ability to predict and classify.

$$(x * y)(t) \stackrel{def}{=} \int_{R^n} x(\tau)y(t - \tau)\,d\tau \tag{2}$$

The function of the fully connected layer is to input the features output from the convolutional layer and the pooling layer into this layer and adjust the weights and biases to obtain accurate classification prediction results. As shown in Figure 20, the fully connected layer of this study is a three-layer $256 \times 128 \times 64$ structure, and the final output is divided into 7 expression classifications (see Figure 15). The dropout layer in the fully connected layer acts to prevent overfitting from occurring during the classification process. During training, the established model will be continuously corrected and reduced the loss trend. The loss of the model can be seen in Figure 21. The loss curve of the model shows a downward trend, which also means that the prediction accuracy can be increased through continuous training. Lastly, the model's accuracy rate is obtained from the following Formula (3). The accuracy definition is derived from the confusion matrix for a given test data set (see Table 1 for details) and the ratio of the number of samples correctly classified by the classification model to the total number of samples (Table 2). This model can reach a high accuracy rate of 87%.

$$\text{Accuracy} = \frac{TN + TP}{TP + TN + FP + FN} \tag{3}$$

TP: correctly identified as positive samples.
TN: correctly identified as negative samples.
FP: falsely identified as positive.

FN: Misidentified as negative sample.

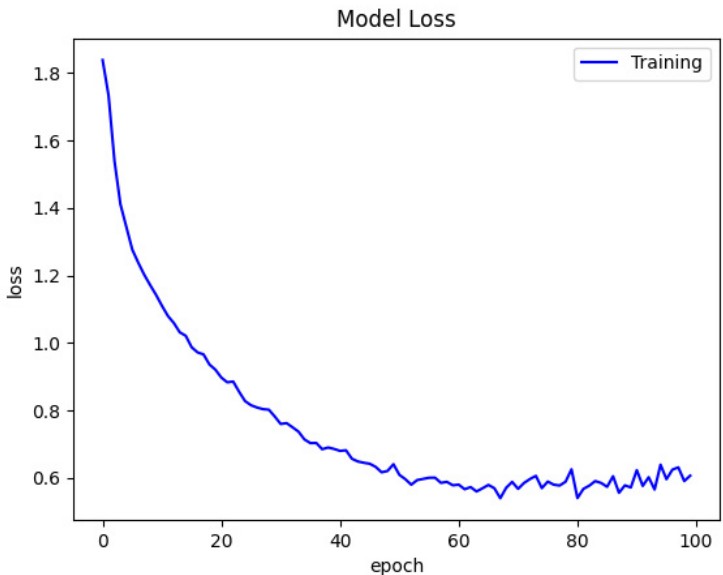

**Figure 21.** Model training loss curve.

**Table 2.** the model prediction results are confused with the positive and negative samples of the data set.

|  | Actual (Positive) | Actual (Negative) |
|---|---|---|
| Predict (Positive) | TP | FP |
| Predict (Negative) | FN | TN |

After the CNN deep learning AI model is trained and integrated with LabVIEW, the detected facial expressions and the judged emotions can be displayed on the human-machine interface. Figures 22 and 23 show the real-time detection of facial emotion by the system. Figure 22 shows the prediction graph of sad expressions, Figure 23 shows the prediction graph of happy expressions, and Figure 24 shows the part of programs used for image processing and sentiment analysis.

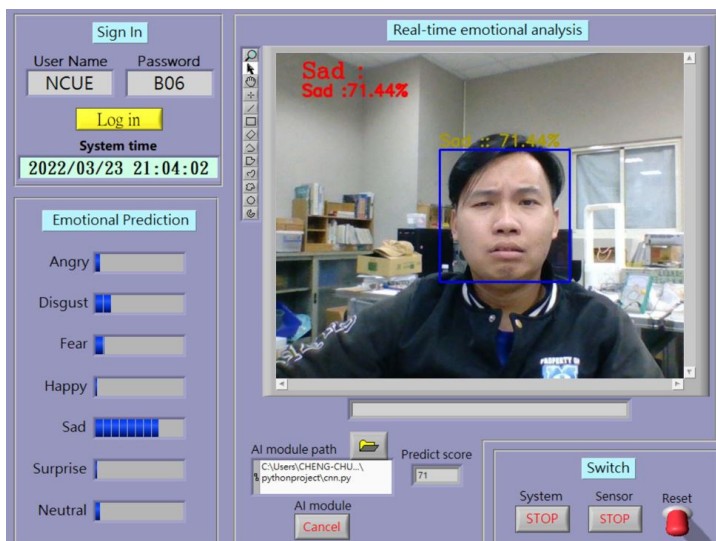

**Figure 22.** Prediction of sad expression.

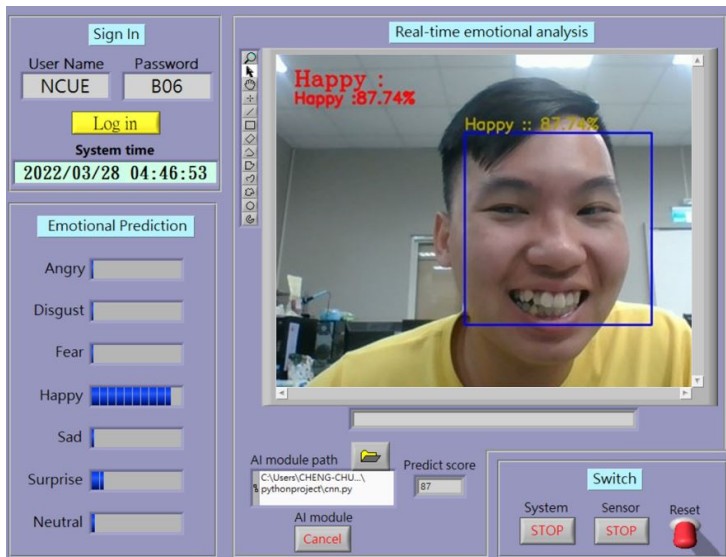

**Figure 23.** Prediction of happy expression.

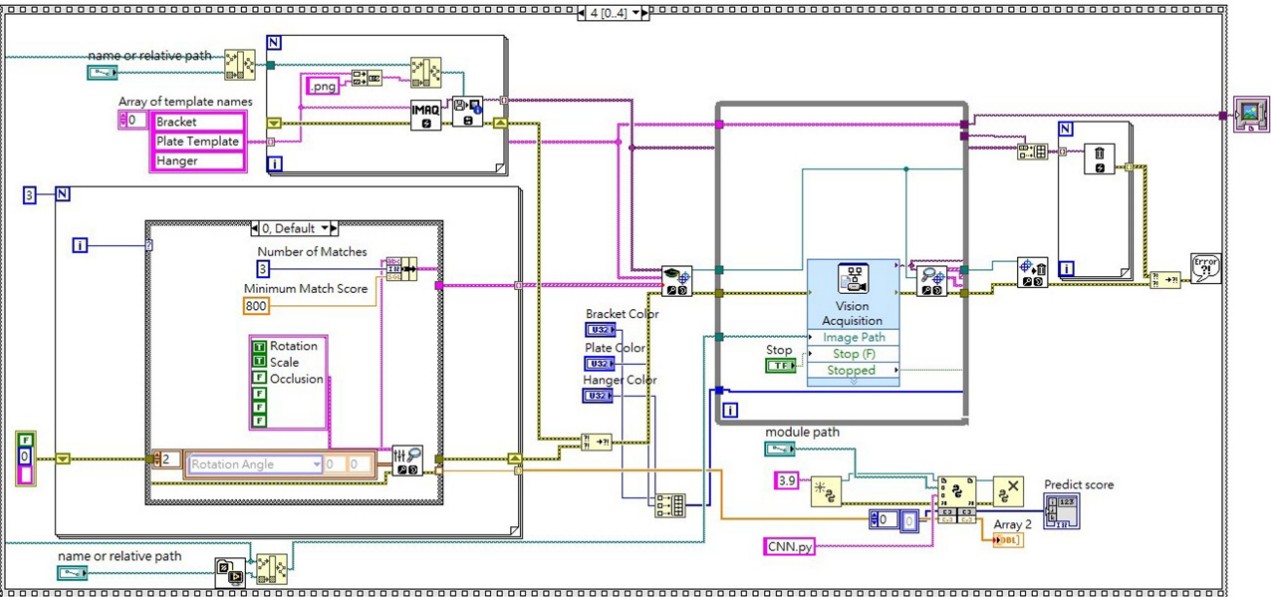

**Figure 24.** Partial program of monitoring and real-time emotion recognition analysis.

This image recognition machine learning system integrates facial emotion recognition and the need of long-term care IoT system to complete an intelligent emotion analysis system of long-term care. The built CNN AI Model can analyze the detected facial expression, and to judge and give the corresponding control to achieve intelligent long-term care operation system, this system can share the work responsibilities and pressures of medical staff and nurses, and can also grasp the psychological feelings of the care recipients in real time, and improve the quality of overall care.

## 5. Conclusions

This study uses AI technology combined with facial image recognition and environmental monitoring to alleviate the problem of medical shortages and the inability of these caregivers to provide comprehensive care in remote rural areas. In this research, LabVIEW is used to construct a smart long-term care system. The software also integrates Python to design a face recognition AI module of convolutional neural network. Furthermore,

this system can also transmit remote data by means of IoT. The established identification database can continuously add training data through the operation of the system, so that the success rate of identification can be improved upwards. The hardware part includes a camera, sensors, and a data capture card for interface use. The integration of software and hardware systems can achieve construction planning of smart long-term care. In practical, this design and construction method will allow this system to add different sensing and control devices and equipment according to long-term care needs. Based on this method, different intelligent long-term care system can be fulfilled.

In similar facial data processing, emotion recognition is subjective information that is difficult to monitor and lacks transparency. It is difficult to draw conclusions when errors or perceptions are doubtful. Therefore, the application proposed in this study is not absolutely correct, but only informative enough to provide warnings and suggestions. In the future, the combination of other detected or measured physiological parameters such as heart rate, oxygen saturation, and body temperature can be used as the next step for more accurate medical applications, and the results can be used to validate the aforementioned facial recognition system to enhance the accuracy of the system.

In this study, the face-to-expression prediction constructed using the LabVIEW platform can work normally in both the hardware and software of the system, but there are still some problems with the test results, including:

1.  There may be more appropriate hyperparameter configurations such as convolutional and fully connected layers, or better depth models may be used to obtain better accuracy.
2.  The amplification of the data volume of the data set, the amount of data in some categories is not sufficient, resulting in the low accuracy of the identification of the category.
3.  Due to national laws and treaty restrictions, more personal identity and health information cannot be added to the research materials, and cannot be disclosed, and the conclusion of the research is easy to be questioned.
4.  The device can use cams with higher resolution and autofocus functions to improve the efficiency of detection and identification, and in the future, it can even obtain more information according to portable electronic devices to achieve a smarter system.

**Author Contributions:** All authors contributed meaningfully to this study. Research topic, K.-C.Y., W.-T.H., T.-Y.C., C.-C.W. and W.-S.H.; methodology, K.-C.Y. and W.-T.H.; software, T.-Y.C. and C.-C.W.; validation, K.-C.Y., W.-T.H., T.-Y.C., C.-C.W. and W.-S.H.; formal analysis, T.-Y.C. and C.-C.W.; investigation, T.-Y.C. and C.-C.W.; resources, K.-C.Y., W.-T.H. and W.-S.H.; data curation, K.-C.Y., W.-T.H. and W.-S.H.; writing—original draft preparation, K.-C.Y., W.-T.H., T.-Y.C., C.-C.W. and W.-S.H.; writing—review and editing, K.-C.Y., W.-T.H., T.-Y.C., C.-C.W. and W.-S.H.; visualization, K.-C.Y., W.-T.H. and W.-S.H.; supervision, K.-C.Y., W.-T.H. and W.-S.H.; project administration, K.-C.Y., W.-T.H. and W.-S.H. All authors have read and agreed to the published version of the manuscript.

**Funding:** This research was partially supported by the Ministry of Science and Technology, Taiwan under the Grant No. MOST 109-2511-H-018-018-MY3.

**Institutional Review Board Statement:** The study was conducted in accordance with the Declaration of Helsinki. The experimental protocol was approved by the National Changhua Normal University Institutional Review Board (IRB).

**Informed Consent Statement:** Informed consent was obtained from all subjects involved in the study.

**Data Availability Statement:** Not applicable.

**Acknowledgments:** This study is grateful for the technical support of the Virtual Instrument Control Center and the Smart Grid Technology and Application Laboratory of National Changhua Normal University.

**Conflicts of Interest:** The authors declare no conflict of interest.

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
