# Peer review of "Establishing an Intelligent Emotion Analysis System for Long-Term Care Application Based on LabVIEW"

_sustainability, doi:10.3390/su14148932_

Round 1
Reviewer 1 Report
This study aimed to build an emotion analysis system for long-term care of the elderly. The topic is interesting. However, the article is overall not well-written, with some critical information missing. The study was also unable to accommodate its aim of caring for the elderly population.
Major comments:
How were the facial images labeled? Were they labeled as percentages of different emotions or a single emotion? As facial expressions can be subjective, who and how many people participated in the labeling process? How are discrepancies handled? Also, it would be helpful if the authors could provide examples of the labeled expressions.
As this AI model is designed for the long-term care of the elderly, it is important to include facial images of the elderly in the training set rather than only young people shown in the examples.
For medical purposes, it is important to detect expressions of pain or agony, which were not included in the labels. Also, it would be more helpful to collect physiological parameters such as heart rate, blood oxygen saturation, and body temperature detected by wearable devices.
Why did the training set use black-and-white images? Color images might provide more information regarding emotion and health status.
How many images were in the training data set? Also, please report the hyperparameters of the CNN model in the main text, such as the number and size of filters and hidden layers.
Was there an external validation of the CNN model, especially in the elderly population? Was the system tested in real-world settings?
An extensive revision of the English language is needed. Some sentences are difficult to understand.
Minor comments:
Line 23-32: This part on the aging population is wordy and can be shortened to 1-2 sentences.
Figure 5, 6 & 7 appear to be repetitive or unnecessary.
Figure 15 provides only general information and can be omitted.
Figure 9 is missing.
It is unclear what Figure 20 represents.
Line 287: It is confusing to use this equation to calculate the accuracy as the output cannot be simply described as positive or negative in this study.
Author Response
Response to Reviewer 1 Comments
This study aimed to build an emotion analysis system for long-term care of the elderly. The topic is interesting. However, the article is overall not well-written, with some critical information missing. The study was also unable to accommodate its aim of caring for the elderly population.
Point 1: How were the facial images labeled? Were they labeled as percentages of different emotions or a single emotion? As facial expressions can be subjective, who and how many people participated in the labeling process? How are discrepancies handled? Also, it would be helpful if the authors could provide examples of the labeled expressions.
Response 1:
(1)Thanks for the reviewer’s suggestion. The authors had been enhanced more literature review [8,9] about facial images in the Session 1 in RED color on rows 48-52 as follows.
In [8], the authors proposed to test the accuracy of successfully predicting seven facial 48 micro-expressions, i.e., happy, sad, angry, scared, surprised, and deceitful-using facial ex-49 pressions for the real-time temptation and aversion recognition dataset (FER-2013), which 50 is the 2013 Facial Expression Recognition Dataset (FER-2013) provided by Kaggle and pre-51 sented at the In-2013 International Conference on Machine Learning (ICML) [9].
(2)According to the classification of literature 9, we collected images of elderly people from Taiwan to the training machine, deleted the data that could not be clearly identified, and used them as training and test sets.
(3)Figure 16 shows the training model incorporates image processing numerical data and manual labels.
(4)Figure 17 shows the image test set before processing.
Point 2: As this AI model is designed for the long-term care of the elderly, it is important to include facial images of the elderly in the training set rather than only young people shown in the examples.
Response 2:
(1)Thanks for the reviewer’s suggestion. The training set also contains images of faces of elderly people. However, it involves research ethics and privacy rights, so only the researcher's experimental map is shown in the example.
Point 3: For medical purposes, it is important to detect expressions of pain or agony, which were not included in the labels. Also, it would be more helpful to collect physiological parameters such as heart rate, blood oxygen saturation, and body temperature detected by wearable devices.
Response 3:
(1)Thanks for the reviewer’s suggestion. Emotion classification has been a topic of debate over the years, and when people are in pain or distress, facial expressions are mostly sad. This study uses single emotion-sadness as the subcategory.
(2)Thanks to the reviewer's suggestion, collecting physiological parameters such as heart rate, blood oxygen saturation, and body temperature detected by wearable devices can indeed improve the accuracy of the model or the overall system. Due to the limitations of rural resources, this study cannot be realized temporarily, and this suggestion will be included in the follow-up research methods.
Point 4: Why did the training set use black-and-white images? Color images might provide more information regarding emotion and health status.
Response 4:
(1)Thanks for the reviewer’s suggestion. Because the use of black and white images in the training process can reduce the complexity of the model and training, in order to improve the recognition efficiency and accuracy, and achieve convenience, speed, and easy maintenance, reducing the original color image to a grayscale image can retain the expression feature points. The instability factor (error) caused by skin color can be reduced.
Point 5: How many images were in the training data set? Also, please report the hyperparameters of the CNN model in the main text, such as the number and size of filters and hidden layers.
Response 5:
(1)Thanks for the reviewer’s suggestion. There are more than 10,000 images in the training data. Because the research is still ongoing, the images in the dataset continue to grow. Rows 283-287 of the manuscript and Figures 18 and 19 show the number and size of filters and hidden layers in training.
Point 6: Was there an external validation of the CNN model, especially in the elderly population? Was the system tested in real-world settings?
Response 6:
(1)Thanks for the reviewer’s suggestion. The goal of this research is model building and system validation. The test model has been validated by training the model and a dataset of facial images of the elderly. Testing in large environments has not yet been conducted.
Point 7: An extensive revision of the English language is needed. Some sentences are difficult to understand.
Response 7:
(1)Thanks for the reviewer’s suggestion. We find a native English speaking colleague to help us to edit the paper.
Point 8: Line 23-32: This part on the aging population is wordy and can be shortened to 1-2 sentences.
Response 8:
(1)Thanks for the reviewer’s suggestions. The authors had been modified and enhanced more critical literature in the Session 1 in RED color on rows 24-42 as follows.
According to the United Nations World Population Prospects 2019 report, the to-tal global population in 2020 will be close to 7.8 billion people, of which 730 million people are over 65 years old, accounting for about 9.3%. It is estimated that by 2050, the number of elderly people will reach 1.5 billion, doubling the growth rate from 2020, accounting for about 16%, showing that the aging of the global population is a common phenomenon [1]. In recent years, the world has faced the double attack of chronic diseases and emerging infectious diseases. Chronic diseases that often occur in the elderly consume a lot of medical resources, and the severe special infectious pneumonia (COVID-19) pandemic has a serious impact on human mental health, showing that chronic diseases and mental diseases have become common problems in modern society. The Industrial Technology Research Institute and Nan Shan Life joint-ly proposed a white paper on long-term care for the elderly in Taiwan in 2021, indi-cating that the elderly industry will combine artificial intelligence and machine learn-ing in the future to transform traditional care into more high-quality and efficient smart care [2]. According to a survey conducted by CommonWealth Magazine in 2021, the average life expectancy of Taiwan’s rural residents is 7 years shorter than that of urban residents, and the lack of medical networks and long-term care is the primary reason. Therefore, it is one of the motives of this study to bridge the gap [3]. The de-velopment of a remote care system has also become an important issue in long-term care for the elderly [4].
Point 9: Figure 5, 6 & 7 appear to be repetitive or unnecessary.
Response 9:
(1)Thanks for the reviewer’s suggestions. The author has corrected it according to your suggestion.
Point 10: Figure 15 provides only general information and can be omitted.
Response 10:
(1)Thanks for the reviewer’s suggestions. The author has corrected it according to your suggestion.
Point 11: Figure 9 is missing.
Response 11:
(1)Thanks for the reviewer’s suggestions. The author has corrected it according to your suggestion.
Point 12: It is unclear what Figure 20 represents.
Response 12:
(1)Thanks for the reviewer’s suggestions. Figure 16(Original Figure 20) shows the training model incorporates image processing numerical data and manual labels. Manuscript rows 250-255 have a detailed description of Figure 16 (Original Figure 20).
Point 13: Line 287: It is confusing to use this equation to calculate the accuracy as the output cannot be simply described as positive or negative in this study.
Response 13:
(1)Thanks for the reviewer’s suggestions. This formula is used to calculate the recognition accuracy of the model, which is calculated from the recognition results of the training set and the test set.
Reviewer 2 Report
First of all, the paper is difficult to read because it does not follow the conventional structures defined in the MDPI author guidelines (Research manuscript sections: Introduction, Materials and Methods, Results, Discussion, Conclusions). I have some important comments about the writing of the article.
- The authors should consider rewriting the abstract to more clearly explain the background, materials and methods, and results.
- The authors should follow the conventional structure as stated in the MDPI Author Guidelines (Research manuscript sections: Introduction, Materials and Methods, Results, Discussion, Conclusions (optional)).
- Better explanations of CNN are needed.
- More reviews of relevant literature are needed.
- You need to explain in more detail the datasets you used to develop the algorithms. I did not clearly explained the dataset you used.
- I did not see exactly how you validated your results. Where was your test set?
- The paper cannot be accepted at this stage because it does not meet the criteria of a scientific paper. The paper needs to be heavily revised.
Author Response
Response to Reviewer 2 Comments
First of all, the paper is difficult to read because it does not follow the conventional structures defined in the MDPI author guidelines (Research manuscript sections: Introduction, Materials and Methods, Results, Discussion, Conclusions). I have some important comments about the writing of the article.
Point 1: The authors should consider rewriting the abstract to more clearly explain the background, materials and methods, and results.
Response 1:
(1)Thanks for the reviewer’s suggestions. The authors had been modified and enhanced more clearly explain the background, materials and methods, and results in the Abstract in RED color on rows 10-20 as follows.
In this study, the authors aimed to implement a smart long-term care system using an AI model that can be integrated with a Laboratory Virtual Instrument Engineering Workbench (LabVIEW) application for sentiment analysis. The input data collected is a database of numerous facial features and environmental variables that have been processed and analyzed; the output decisions are the corresponding controls for sentiment analysis and prediction. Convolutional Neural Network (CNN) is used to deal with the complex process of deep learning. After the con-volutional layer simplifies the processing of the image matrix, the results are computed by the fully connected layer. In addition, the MLP model embedded in LabVIEW for data analysis and predictive control [14] is constructed to predict the corresponding control of emotional and envi-ronmental variables. Moreover, LabVIEW is used to design sensor components, data displays, and control interfaces. Remote sensing and control is achieved by using LabVIEW’s built-in web publishing tools.
Point 2: The authors should follow the conventional structure as stated in the MDPI Author Guidelines (Research manuscript sections: Introduction, Materials and Methods, Results, Discussion, Conclusions (optional)).
Response 2:
(1)Thanks for the reviewer’s suggestions. Because the author's manuscript is biased towards papers related to engineering technology, and the other four reviewers have no suggestions in this regard. Two reviewer’s even stated that the paper is logically structured and the ideas are clearly presented. Therefore, the part of the chapter name is kept as it is, but many contents and important documents are supplemented.
Point 3: Better explanations of CNN are needed.
Response 3:
(1)Thanks for the reviewer’s suggestion. The authors had been enhanced more literature review [35] about CNN in the Session 2 in RED color on rows 93-101 as follows.
The most beneficial aspect of CNNs is reducing the number of parameters in ANN. This achievement has prompted both researchers and developers to approach larger models in order to solve complex tasks, which was not possible with classic ANNs;. The most important assumption about problems that are solved by CNN should not have features which are spatially dependent. In other words, for example, in a face de-tection application, we do not need to pay attention to where the faces are located in the images[35]. Therefore, this research builds an AI model for real-time image sentiment analysis based on the CNN model.
Point 4: More reviews of relevant literature are needed.
Response 4:
(1)Thanks for the reviewer’s suggestion. More recent studies in related fields have been added to the literature section as reference evidence for this study, the limitations of the study have been formulated, more research has been conducted on the literature, and differences have been revised.
Point 5: You need to explain in more detail the datasets you used to develop the algorithms. I did not clearly explained the dataset you used.
Response 5:
(1)Thanks for the reviewer’s suggestion. The authors had been enhanced more literature review [8,9] about facial images in the Session 1 in RED color on rows 48-52 as follows.
In [8], the authors proposed to test the accuracy of successfully predicting seven facial 48 micro-expressions, i.e., happy, sad, angry, scared, surprised, and deceitful-using facial ex-49 pressions for the real-time temptation and aversion recognition dataset (FER-2013), which 50 is the 2013 Facial Expression Recognition Dataset (FER-2013) provided by Kaggle and pre-51 sented at the In-2013 International Conference on Machine Learning (ICML) [9].
(2)According to the classification of literature 9, we collected images of elderly people from Taiwan to the training machine, deleted the data that could not be clearly identified, and used them as training and test sets.
(3)Figure 16 shows the training model incorporates image processing numerical data and manual labels.
(4)Figure 17 shows the image test set before processing.
Point 6: I did not see exactly how you validated your results. Where was your test set?
Response 6:
(1)Thanks for the reviewer’s suggestion. Because the main goal of this research is to be able to be used in remote villages, and it has the characteristics of fast modeling, cheap price, and easy maintenance of the system. Therefore, this study currently focuses on system construction, firstly verifying the feasibility of the model and system, and has not yet verified the experimental results in the actual environment. This study describes the model validation steps in rows 291-298, and tests the results through the calculation formula in row 300 and the model loss in Figure 20. Figure 17 shows part of the test set data after labeling. The test set also uses facial images of the elderly for training, following the data set explanation above.
Point 7: The paper cannot be accepted at this stage because it does not meet the criteria of a scientific paper. The paper needs to be heavily revised.
Response 7:
(1)Thanks for the reviewer’s suggestions. The author has corrected it according to your suggestion.
Reviewer 3 Report
The authors propose Establishing an Intelligent Emotion Analysis System for Long- term Care Application based on LabVIEW. I have some concerns and my suggestions are listed below:
1. In the abstract, the contribution is not properly explained. The motivation for the paper does not exist. The input was not explained in a way that was clear and understandable. The abstract section should be modified to emphasize the work's major point.
2. The authors should focus on the study's main issue in the opening section and include a Literature Review in the form of tables to highlight research gaps and innovations.
3. To synthesize the overall information from the related literature and to present the paper's main points, the authors should use a table. This table may be useful in identifying the discrepancies between the proposed model and previous work.
4. The authors did not evaluate the advantages and disadvantages of the connected works. In the related work section, please explain how their study differs from that of others. What do they have that the others don't? What makes them better, and how do they do it?
5. Experimental Results, validation, and comparison to other approaches should be improved. More discussions and more analysis are needed.
6. It is very important to introduce the computational complexity of Lab-VIEW model.
7. Nonparametric tests should be performed for experiments.
Author Response
Response to Reviewer 3 Comments
The authors propose Establishing an Intelligent Emotion Analysis System for Long- term Care Application based on LabVIEW. I have some concerns and my suggestions are listed below:
Point 1: In the abstract, the contribution is not properly explained. The motivation for the paper does not exist. The input was not explained in a way that was clear and understandable. The abstract section should be modified to emphasize the work's major point.
Response 1:
(1)Thanks for the reviewer’s suggestions. The authors had been modified and enhanced more clearly explain the background, materials, methods and contribution, and results in the Abstract in RED color on rows 10-20 as follows.
In this study, the authors aimed to implement a smart long-term care system using an AI model that can be integrated with a Laboratory Virtual Instrument Engineering Workbench (LabVIEW) application for sentiment analysis. The input data collected is a database of numerous facial features and environmental variables that have been processed and analyzed; the output decisions are the corresponding controls for sentiment analysis and prediction. Convolutional Neural Network (CNN) is used to deal with the complex process of deep learning. After the con-volutional layer simplifies the processing of the image matrix, the results are computed by the fully connected layer. In addition, the MLP model embedded in LabVIEW for data analysis and predictive control [14] is constructed to predict the corresponding control of emotional and envi-ronmental variables. Moreover, LabVIEW is used to design sensor components, data displays, and control interfaces. Remote sensing and control is achieved by using LabVIEW’s built-in web publishing tools.
Point 2: The authors should focus on the study's main issue in the opening section and include a Literature Review in the form of tables to highlight research gaps and innovations.
Response 2:
(1)Thanks for the reviewer’s suggestion. More recent studies in related fields have been added to the literature section as reference evidence for this study, the limitations of the study have been formulated, more research has been conducted on the literature, and differences have been revised.
Point 3: To synthesize the overall information from the related literature and to present the paper's main points, the authors should use a table. This table may be useful in identifying the discrepancies between the proposed model and previous work.
Response 3:
(1)Thanks for the reviewer’s suggestion. More recent studies in related fields have been added to the literature section as reference evidence for this study, the limitations of the study have been formulated, more research has been conducted on the literature, and differences have been revised.
Point 4: The authors did not evaluate the advantages and disadvantages of the connected works. In the related work section, please explain how their study differs from that of others. What do they have that the others don't? What makes them better, and how do they do it?
Response 4:
(1)Thanks for the reviewer’s suggestion. In this study, the author refers to the advantages of many commodities to develop a new system for remote rural areas or areas with insufficient medical resources.
Point 5: Experimental Results, validation, and comparison to other approaches should be improved. More discussions and more analysis are needed.
Response 5:
(1)Thanks for the reviewer’s suggestions. The authors had been modified and enhanced more results, validation, and comparison, and results in the Conclusion in RED color on rows 331-333, 344-351 as follows.
This study uses AI technology combined with facial image recognition and environmental monitoring to alleviate the problem of medical shortages and the inability of these caregivers to provide comprehensive care in remote rural areas.
In similar facial data processing, emotion recognition is subjective information that is difficult to monitor and lacks transparency. It is difficult to draw conclusions when errors or perceptions are doubtful. Therefore, the application proposed in this study is not absolutely correct, but only informative enough to provide warnings and suggestions. In the future, the combination of other detected or measured physiological parameters such as heart rate, oxygen saturation, and body temperature can be used as the next step for more accurate medical applications, and the results can be used to validate the aforementioned facial recognition system to enhance the accuracy of the system.
Point 6: It is very important to introduce the computational complexity of Lab-VIEW model.
Response 6:
(1)Thanks for the reviewer’s suggestions. The author has added more introduction to the integration complexity of LabVIEW and Ai models, sensors, etc in RED color on rows 190-200 and Figure 5, 9 as follows.
In the AI model design, Python is used to construct the required AI modules, train and test them first, and then verify the feasibility of the model. The part using the programming approach is shown in Figure 9. The AI model (Python CNN archi-tecture) is embedded in the system as an API using the Labview Python node. The sys-tem receives the detected images instantly on the human-machine interface, and the trained model is used to verify the facial expressions and perform predictive analysis at the back-end of the system. In addition, the detected image data is further trans-ferred back to the new training set for subsequent model validation. The obtained ex-pressions are quantified by Labview for the prediction scores of the sentiment, as shown in Figure 8. This section corresponds to A in Block Figure 4, and the results of these operations will be used for various control decisions in the long-term care envi-ronment.
Point 7: Nonparametric tests should be performed for experiments.
Response 7:
(1)Thanks for the reviewer’s suggestion. Because the main goal of this research is to be able to be used in remote villages, and it has the characteristics of fast modeling, cheap price, and easy maintenance of the system. Therefore, this study currently focuses on system construction, firstly verifying the feasibility of the model and system, and has not yet verified the experimental results in the actual environment. This study describes the model validation steps in rows 291-298, and tests the results through the calculation formula in row 300 and the model loss in Figure 20. Figure 17 shows part of the test set data after labeling. The test set also uses facial images of the elderly for training, following the data set explanation above.
Reviewer 4 Report
The manuscript develops a topic of great interest. The authors aim to construct an AI emotion recognition model and a long- term care medical system with remote monitoring functions. Such a model could have great practical applicability.
The paper is logically structured and the ideas are clearly presented.
However, I think it is worthwhile to improve the work. Thus, in the paragraph “Main Results”, the authors should compare the results obtained with other results from the literature and explain the differences.
The „Conclusions” paragraph should also be extended. In this paragraph, the authors should emphasize the main results obtained, their originality and usefulness, and the limitations of the research.
Author Response
Response to Reviewer 4 Comments
The manuscript develops a topic of great interest. The authors aim to construct an AI emotion recognition model and a long- term care medical system with remote monitoring functions. Such a model could have great practical applicability.
The paper is logically structured and the ideas are clearly presented.
Point 1: However, I think it is worthwhile to improve the work. Thus, in the paragraph “Main Results”, the authors should compare the results obtained with other results from the literature and explain the differences.
Response 1:
(1)Thanks for the reviewer’s suggestions. Because facial emotion recognition is subjective and can be deceived.On the application side, the system is used as a premise for faster and more accurate to assist in handling and early warning of emergency situations, reducing the pressure on caregivers.Combined with the monitoring of the environment, this system can eliminate environmental problems at the first time, and then provide assistance according to the physiological conditions of the specific care recipients.
Point 2: The „Conclusions” paragraph should also be extended. In this paragraph, the authors should emphasize the main results obtained, their originality and usefulness, and the limitations of the research.
Response 2:
(1)Thanks for the reviewer’s suggestions. The authors had been modified and enhanced more the main results obtained, their originality, usefulness, and the limitations of the research in the Conclusion in RED color on rows 331-333, 344-351 as follows.
This study uses AI technology combined with facial image recognition and environmental monitoring to alleviate the problem of medical shortages and the inability of these caregivers to provide comprehensive care in remote rural areas.
In similar facial data processing, emotion recognition is subjective information that is difficult to monitor and lacks transparency. It is difficult to draw conclusions when errors or perceptions are doubtful. Therefore, the application proposed in this study is not absolutely correct, but only informative enough to provide warnings and suggestions. In the future, the combination of other detected or measured physiological parameters such as heart rate, oxygen saturation, and body temperature can be used as the next step for more accurate medical applications, and the results can be used to validate the aforementioned facial recognition system to enhance the accuracy of the system.
Reviewer 5 Report
This paper described a implement a smart long-term care system using an AI model using LABView, and experiments showed that the developed system can fulfill intelligent long-term care mission. Overall, the paper is well-organized and easy to follow. The work is good, but still has some significant weaknesses.
1.How to distinguish the current work from any other studies regarding long-term care system using an AI model?
2.What is special in long-term care system using an AI model design? The CNN model for face recognition or emotion analysis is a mature technology and implmented in existing remote care system. The requirements of developed system have tested and validated with ongoing medical care units?
3.The proposed method should be compared with the state-of-the-art face recognition algorithms. Suggest the open face recognition databases to validate the experimental results.
4. Some references need to be updated with relevant and recent papers focused on the fields.
Author Response
Response to Reviewer 5 Comments
This paper described a implement a smart long-term care system using an AI model using LabVIEW, and experiments showed that the developed system can fulfil intelligent long-term care mission. Overall, the paper is well-organized and easy to follow. The work is good, but still has some significant weaknesses.
Point 1: 1.How to distinguish the current work from any other studies regarding long-term care system using an AI model?
Response 1:
(1)Thanks for the reviewer’s suggestions. This research mainly focuses on factors such as information gap in remote villages, insufficient medical care, etc., and builds a graphical interface and program structure that is easy to maintain. It uses fast and convenient methods and low-cost simple equipment to make preliminary judgments, so as to timely detect the situation and give early warning to caregivers.
Point 2: What is special in long-term care system using an AI model design? The CNN model for face recognition or emotion analysis is a mature technology and implemented in existing remote care system. The requirements of developed system have tested and validated with ongoing medical care units?
Response 2:
(1)Thanks for the reviewer’s suggestions. The design of this AI model is to use simple equipment to establish a long-term care system, trying to greatly reduce the burden of caregivers in the case of insufficient human resources. At present, the research is in the stage of building a model and validating system, and has passed the review of the ethics committee of the academic institution, but has not yet been tested and validated with the health care unit.
Point 3: The proposed method should be compared with the state-of-the-art face recognition algorithms. Suggest the open face recognition databases to validate the experimental results.
Response 3:
(1)Thanks for the reviewer’s suggestion. The authors had been enhanced more literature review [8,9] about facial images in the Session 1 in RED color on rows 48-52 as follows.
In [8], the authors proposed to test the accuracy of successfully predicting seven facial 48 micro-expressions, i.e., happy, sad, angry, scared, surprised, and deceitful-using facial ex-49 pressions for the real-time temptation and aversion recognition dataset (FER-2013), which 50 is the 2013 Facial Expression Recognition Dataset (FER-2013) provided by Kaggle and pre-51 sented at the In-2013 International Conference on Machine Learning (ICML) [9].
(2) Due to the norms of the Research Ethics Committee and Personal Information Law, we are unable to provide recognition database involves the identity information of the subjects.
Point 4: Some references need to be updated with relevant and recent papers focused on the fields.
Response 4:
(1)Thanks for the reviewer’s suggestion. More recent studies in related fields have been added to the literature section as reference evidence for this study, the limitations of the study have been formulated, more research has been conducted on the literature, and differences have been revised.
Round 2
Reviewer 1 Report
I want to thank the authors for making the efforts to address my concerns. However, some of the questions are still not well answered. The authors failed to provide some critical information, which diminishes the reliability of the study.
1. How many images of elderly people were included in the training process? Please provide the exact number and percentage in the manuscript.
2. Please give the names of the people who implement the labeling process. Please describe in detail how discrepancies were handled.
3. Please provide the hyperparameters of the CNN model in the text. Also, please add necessary footnotes to the figures so that an average reader of Sustainability can easily understand them.
4. Please add a paragraph discussing the limitations to this study, which should incorporate the concerns the authors fail to address and the improvement they suggest for future studies.
5. Line 300: I still do not understand why this equation is relevant. Please define what is positive/negative in this scenario.
Author Response
Response to Reviewer 1 Comments
I want to thank the authors for making the efforts to address my concerns. However, some of the questions are still not well answered. The authors failed to provide some critical information, which diminishes the reliability of the study.
Point 1: How many images of elderly people were included in the training process? Please provide the exact number and percentage in the manuscript.
Response 1:
(1)Thanks for the reviewer’s suggestions. The author has added more Interpretation of the amount of data used in the training process in RED color on rows 279-282 and Table 1 as follows.
In this study, the data classification of the FER-2013 dataset was referred to, and the dataset was labeled according to seven different classifications of happy, anger, sad-ness (pain), fear, disgust, surprise, and contempt, and the number of data in the da-taset is shown in Table 1.
Table 1. Statistical Table of Testing and Validation Data of Research Datasets.
Micro-expression |
Validation data |
Training data |
||||
(classification) |
elder |
others |
total |
elder |
others |
others |
happy |
345 |
1248 |
1593 |
2315 |
6784 |
9099 |
anger |
244 |
976 |
1220 |
1042 |
4253 |
5295 |
sadness |
287 |
881 |
1168 |
2151 |
5382 |
7533 |
fear |
193 |
829 |
1022 |
1097 |
4721 |
5818 |
disgust |
43 |
153 |
196 |
282 |
805 |
1087 |
surprise |
218 |
716 |
934 |
1126 |
3686 |
4812 |
contempt |
284 |
996 |
1280 |
1972 |
5819 |
7791 |
1614 |
5799 |
7413 |
9985 |
31450 |
41435 |
Point 2: Please give the names of the people who implement the labeling process. Please describe in detail how discrepancies were handled.
Response 2:
(1)Thanks for the reviewer’s suggestions. This study has been approved by the research institution ethics review board in 2021. Relevant supporting documents have been attached with the manuscript submission. Therefore, in order to comply with research ethics committee regulations, we cannot disclose the names of those who implemented the labeling process.
This study is based on the classification features in the literature [39] to do differential processing. The author has added more Interpretation of the processing methods and definitions of the appearance characteristics of the elderly and young, in RED color on rows 164-186 as follows.
In the emotion recognition system, after the image captures the facial features, the points of difference are distinguished from them. For example: eyebrow contour, eye contour, pupil, nostrils, mouth contour and mouth center, organized into seven kinds of micro-expression mark judgment classification criteria. As follows:
- happy: the cheek muscles rise, the corners of the mouth rise, the corners of the mouth pull back, the eyebrows are flat, and the eyes become smaller;
- sadness: upper eyelid drooping, dilated pupils, corners of the mouth pulling down, cheeks pulling down, eyebrows locked deeply;
- anger: enlarged nostrils, enlarged eyes;
- disgust: raise your nose and raise your mouth;
- fear: the middle of the eyebrows is crowded together;
- surprise: the mouth is slightly open, the pupils are dilated, and the eyebrows are raised;
- contempt: The corners of the lips tighten and lift only one side of the face, and one eyebrow rises;
Figure 4. The image on the left is an example of a sad feature map; in the middle and to the right are example diagrams of the characteristics of the concept of different ages.
From Figure 4, the same emotion under the feature points and age are not much different, and after the image is grayscale processed, the skin color will not have a big impact, and the AI model can be trained to identify according to the feature points of these emotions.
Point 3: Please provide the hyperparameters of the CNN model in the text. Also, please add necessary footnotes to the figures so that an average reader of Sustainability can easily understand them.
Response 3:
(1)Thanks for the reviewer’s suggestions. The author has added more introduction to the integration complexity of the description of the hyperparameters of the CNN model, including the description of the number of features, the number of convolution and pooling layers, and the hidden neurons of the fully connected layer, etc in RED color on rows 279-282, 289, 296-298, 340-348, 353, 360-365 and Figure 18-19 and Table 1 as follows.
In this study, the data classification of the FER-2013 dataset was referred to, and the dataset was labeled according to seven different classifications of happy, anger, sad-ness (pain), fear, disgust, surprise, and contempt, and the number of data in the da-taset is shown in Table 1.
Step 4: Analyze model loss and accuracy through training and test sets.
Table 1. Statistical Table of Testing and Validation Data of Research Datasets.
Micro-expression |
Validation data |
Training data |
||||
(classification) |
elder |
others |
total |
elder |
others |
others |
happy |
345 |
1248 |
1593 |
2315 |
6784 |
9099 |
anger |
244 |
976 |
1220 |
1042 |
4253 |
5295 |
sadness |
287 |
881 |
1168 |
2151 |
5382 |
7533 |
fear |
193 |
829 |
1022 |
1097 |
4721 |
5818 |
disgust |
43 |
153 |
196 |
282 |
805 |
1087 |
surprise |
218 |
716 |
934 |
1126 |
3686 |
4812 |
contempt |
284 |
996 |
1280 |
1972 |
5819 |
7791 |
1614 |
5799 |
7413 |
9985 |
31450 |
41435 |
Such as Figure 19 four-block red box, after the 4-layer convolution and pooling process designed in this study, the author refers to the FER-2013 database to convert the collected full-color images into a first-order matrix image,black and white is more friendly to the model. After the generalized convolution operation, such as formula (1), The function x and y are measurable function defined on . The convolution of x and y is denoted as . It is the integral of the product of one of the functions after inversion and translation and the product of the other function, which is a pair A function of the amount of translation, the number of features obtained after the cumulative operation are displayed in the param column.
As shown in Figure 20, the fully connected layer of this study is a three-layer 256*128*64 structure, and the final output is divided into 7 expression classifications (see Figure 15). The dropout layer in the fully connected layer acts to prevent overfit-ting from occurring during the classification process. During training, the established model will be continuously corrected and reduced the loss trend.
Point 4: Please add a paragraph discussing the limitations to this study, which should incorporate the concerns the authors fail to address and the improvement they suggest for future studies.
Response 4:
(1)Thanks for the reviewer’s suggestions. The author has added more instructions of current research issues and future improvements in RED color on rows 430-444 as follows.
In this study, the face-to-expression prediction constructed using the LabVIEW platform can work normally in both the hardware and software of the system, but there are still some problems with the test results, including:
- There may be more appropriate hyperparameter configurations such as convolutional and fully connected layers, or better depth models may be used to obtain better accuracy.
- The amplification of the data volume of the data set, the amount of data in some categories is not sufficient, resulting in the low accuracy of the identification of the category.
- Due to national laws and treaty restrictions, more personal identity and health information cannot be added to the research materials, and cannot be disclosed, and the conclusion of the research is easy to be questioned.
- The device can use cams with higher resolution and autofocus functions to improve the efficiency of detection and identification, and in the future, it can even obtain more information according to portable electronic devices to achieve a smarter system.
Point 5: Line 300: I still do not understand why this equation is relevant. Please define what is positive/negative in this scenario.
Response 5:
(1)Thanks for the reviewer’s suggestions. The author has added more instructions of relevant in RED color on rows 367-370, 379-381 and Table 2 as follows.
Lastly, the model’s accuracy rate is obtained from the following formula (2). The accuracy definition is derived from the confusion matrix for a given test data set (see Table 1 for details) and the ratio of the number of samples correctly classified by the classification model to the total number of samples (Table 2). This model can reach a high accuracy rate of 87%.
Table 2. the model prediction results are confused with the positive and negative samples of the data set.
|
Actual (Positive) |
Actual (Negative) |
Predict (Positive) |
TP |
FP |
Predict (Negative) |
FN |
TN |

Reviewer 2 Report
- First of all, other reviewers comments have nothing to do with mine.
- In the abstract, please fix abbreviations and remove citation.
- You have taken a lot of sentences from your previous paper: https://www.hindawi.com/journals/mpe/2021/7572818/. This is literally not okay. Why don't you try to paraphrase a bit? You previous version has a Turnitin's similarity score of 32%! You current revision has a Turnitin's similarity score of 35%! Still unacceptable practice.
Author Response
Response to Reviewer 2 Comments
First of all, other reviewers comments have nothing to do with mine.
Point 1: In the abstract, please fix abbreviations and remove citation.
Response 1:
(1) Thanks for the reviewer’s suggestions. The authors have removed citations and corrected abbreviations.
Point 2: You have taken a lot of sentences from your previous paper: https://www.hindawi.com/journals/mpe/2021/7572818/. This is literally not okay. Why don't you try to paraphrase a bit? You previous version has a Turnitin's similarity score of 32%! You current revision has a Turnitin's similarity score of 35%! Still unacceptable practice.
Response 2:
(1)Thanks for the reviewer’s suggestions. This study has been approved by the research institution ethics review board in 2021. Relevant supporting documents have been attached with the manuscript submission.
Regarding reviewers' concerns. We attach great importance to it, and have inquired about relevant laws and regulations (Key points for handling and reviewing academic ethics cases of the Ministry of Science and Technology; principles for handling academic ethics cases in colleges and universities of the Ministry of Education). And invited members of the ethics committee to review.After the final review by the members of the ethics committee, the two articles were judged to be no problem.
This paper and the previous one: https://www.hindawi.com/journals/mpe/2021/7572818/ are advanced research and extended applications in the same comprehensive large-scale project. For the similarity problem, after ethics committee member checking there is no problem. But the author still adds more explanations and differences in the text to address the reviewers' concerns. The latest revised manuscript was compared by Turnitin software and showed a similarity of 13%.
There are a total of 5 reviewers for this article. The remaining four reviewers gave the affirmation of "The author provided a fairly detailed answer to the review comments. I have no further questions." on Round 2. We are convinced that this is the result of a rigorous and objective review.
Each reviewer's comments are important to the authors. The author will try to make changes according to the suggestions of each reviewer, and will not ignore any valuable comments. Thank you for your comments to make this research paper better.

Reviewer 5 Report
The authors have provided quite elaborate answers to the reviewing comments. I am glad to see the tuning of the position of the paper be more clear. I don't have further questions.
Author Response
Response to Reviewer 5 Comments
Point 1: The authors have provided quite elaborate answers to the reviewing comments. I am glad to see the tuning of the position of the paper be more clear. I don't have further questions.
Response 1:
(1) Thank you for your comments to make this research paper better.

Round 3
Reviewer 1 Report
I want to thank the authors for their detailed answers to my questions. All my concerns have been addressed adequately.
Author Response
Point 1: I want to thank the authors for their detailed answers to my questions. All my concerns have been addressed adequately.
Response 1:
(1) Thank you for your comments to make this research paper better.
